# Effect of an 11-Week Repeated Maximal Lumbar Movement with Controlled Breathing on Lumbar Sagittal Range of Motion in Elite Swimmers: A Randomized Clinical Trial

**DOI:** 10.3390/healthcare13050457

**Published:** 2025-02-20

**Authors:** Mónica Solana-Tramunt, Ana Bofill-Ródenas, Josep Cabedo, Alesander Badiola-Zabala, Myriam Guerra-Balic

**Affiliations:** 1Facultat de Psicologia, Ciències de l’Educació i de l’Esport Blanquerna, Universitat Ramon Llull, 08022 Barcelona, Spain; josepcs@blanquerna.url.edu (J.C.); myriamgb@blanquerna.url.edu (M.G.-B.); 2Institut Nacional d’Educació Física de Catalunya (INEFC), Universitat de Barcelona, 08038 Barcelona, Spain; abofill@gencat.cat; 3Facultat de Ciències de la Salut Blanquerna, Universitat Ramon Llull, 08025 Barcelona, Spain; alesanderbz@blanquerna.url.edu

**Keywords:** lumbar mobility, undulatory underwater swimming, high performance, conscious movement, dynamic stretching, core training, motor control

## Abstract

Lumbar range of motion (ROM) is essential to develop effective movements during the underwater undulatory swimming technique. Core exercises are used to improve the strength of the muscles that participate in that technique, and variations in sensory input and attentional focus may modulate neuromuscular responses and impact training outcomes. The aim of this study was to investigate the impact of an 11-week program of repeated maximal lumbar movements with closed eyes and without focused attention on lumbar sagittal ROM in elite swimmers versus executing them solely with proper exercise technique with controlled breathing. Methods: A sample of 57 professional swimmers, including 34 males (20.2 ± 4.2 years) and 23 females (20.7 ± 3.3 yrs), volunteered to complete this study. They were randomly divided into two experimental groups (EG1 and EG2) and one control group (CG). All subjects underwent the same type of training program in parallel with the EG intervention. EG1 and EG2 performed three sets of ten repetitions of lumbar flexion and extension exercises at breathing pace, 6 days a week for 11 weeks. EG1 performed the core workout with closed eyes and focused attention on the lumbar movement, while EG2 only followed the technique of the exercises at a controlled breathing pace. Lumbar flexion (F), extension (E), and total ROM (TROM) were assessed by an electrogoniometer in a seated, relaxed position over a Swiss ball. Results: Repeated measures ANOVA showed significant differences in the multivariate profiles across groups and over time. F (8, 48) = 3.495, *p* = 0.002. EG1 had non-significant increases in lumbar ROM, EG2 had significant increases in TROM and extension ROM, and CG had no changes. Conclusions: The results suggest that repeating maximal lumbar movement at a controlled breathing pace, with opened eyes and non-focusing attention on the movement, increases lumbar ROM in the sagittal plane.

## 1. Introduction

Lumbar range of motion (ROM) has been studied to determine changes that could cause core instability, low back pain, or other lumbar disorders [1,2,3,4,5]. Lumbar ROM is essential to develop effective movements during both daily living and sports activities, and it has been an important factor in low back pain, especially when the “Neutral Zone” (the region of intervertebral motion around neutral posture where little resistance is offered by the passive spinal column) is increased [6,7]. A normal lumbar range of motion (ROM) enables the body to respond better to external forces and helps prevent most lumbar injuries and pain. A significant reduction in lumbar ROM, particularly in the sagittal plane, has been observed in individuals with low back pain. Moreover, those with low back pain tend to experience a general decrease in lumbar ROM [5,8].

The core area is considered as one of the focuses in most strength and conditioning programs in elite swimmers, as it has been proposed that the core muscles contract during swimming to decrease form resistance or drag on a swimmer’s body to increase speed [9,10,11,12]. In swimming, the maintenance of posture, balance, and alignment is believed to be critical to maximizing propulsion and reducing drag [9,13,14,15].

Recently, it was reported that differences among high-level swimmers in terms of performance stem from their ability to perform high-speed undulatory underwater swimming (UUS). Undulatory underwater swimming efficiency depends on the lumbar range of movement, core stiffness, and the swimmers’ ability to control and develop maximal lumbar flexion and extension velocity, followed by the inertial movement of the hips, knees, and ankles [14,15,16,17,18]. UUS is used after diving starts and turns in competitive swimming, and the technique can help maintain the high swimming velocity obtained by pushing off the start block or wall. Several studies have reported that the swimming velocity during the underwater phase is related to the total performance of the start and the turn phases [13,14,15,16,19]. Therefore, improvements in the performance of the underwater dolphin kick can reduce times at the start or turns and improve overall performance. Several previous studies reported relationships between kinematic parameters and swimming performance during the underwater dolphin kick. Other studies were developed to analyze the motor patterns of UUS and found that flexor and extensor muscles in the trunk, thigh, and leg are required to contract alternately during dolphin kicking. More specifically, measurements were taken for the following core muscles of each swimmer: rectus abdominis (RA), internal oblique (IO), erector spinae (ES), and multifidus (MF). The study by Matsuura et al. [14] analyzed muscular synergies at three moments of UUS: the transition from the upward kick to the downward kick, the downward kick, and the upward kick. In the UUS of elite swimmers, both the upward and downward kicks followed the activation pattern of trunk muscles involved in the pelvic forward–backward tilt movement, along with the activation of muscles in the lower limbs [14,15].

Core control exercises play a crucial role in enhancing postural stability, movement efficiency, and injury prevention across various athletic and rehabilitative contexts. These exercises target deep and superficial muscles, particularly the transversus abdominis, multifidus, and obliques, which are essential for maintaining spinal alignment and generating force during dynamic movements [9,10,11,20,21,22]. There is strong evidence suggesting the large beneficial influence of strengthening on posture. However, the underlying mechanisms are a matter of debate. Surprisingly, there is a lack of conclusive research on resistance training-induced changes in muscle passive mechanical properties [23]. Research suggests that sensory inputs, such as vision and proprioception, significantly influence motor control during core exercises. Closing the eyes during movement training reduces visual dependence, forcing the body to rely more on proprioceptive feedback, particularly from muscle spindles. Muscle spindles, which detect changes in muscle length and tension, become more engaged when visual input is removed, potentially enhancing neuromuscular coordination and proprioceptive sensitivity. Additionally, consciously focusing attention on muscle activation during core exercises can improve motor learning and enhance muscle recruitment, leading to better control and efficiency. This principle is widely applied in rehabilitation and performance training to optimize motor patterns and improve functional stability [24,25].

Core workouts are essential components of swimming strength and conditioning programs and are often incorporated into dryland warm-ups in swimming practice. The significant improvement in core muscular function contributes to enhanced swimming performance [9,10,11]. Moreover, current swimming dry-land warm-ups include mobility exercises, as trunk and shoulder mobility are crucial for technical requirements in swimming strokes [11,26,27]. Nevertheless, to our knowledge, the effect of core exercises on lumbar range of motion remains unclear [28,29,30]. Investigating the differences between performing core exercises with closed eyes and without focused attention versus executing them solely with proper exercise technique is essential for understanding the role of sensory and cognitive factors in motor control and neuromuscular adaptations [25,31,32]. Core training is widely utilized in both athletic performance and rehabilitation contexts to enhance stability, strength, and injury prevention. However, variations in sensory input and attentional focus may modulate neuromuscular responses and impact training outcomes. Therefore, this study aimed to investigate the impact of an 11-week program of repeated maximal lumbar movements with closed eyes and without focused attention on the lumbar sagittal range of motion in elite swimmers versus executing them solely with the proper exercise technique with controlled breathing. We hypothesized that these training programs would lead to a significant increase in swimmers’ total lumbar range of motion compared to the control group.

## 2. Materials and Methods

### 2.1. Study Design

We conducted a randomized clinical trial according to the CONSORT checklist [33]. The arms of this study and its flowchart are shown in Figure 1. Informed consent was obtained from all participants, and all procedures were conducted according to the Declaration of Helsinki. The research protocol was approved by the institutional Human Research Ethics Committee of Ramon Llull University, Spain. This study was registered on the Trial registration Current Controlled Trials website at www.ClinicalTrials.gov (accessed on 4 December 2024) (NCT06747702).

### 2.2. Study Population

GPOWER v3.1 software (Bonn FRG, University of Bonn, Department of Psychology) was used to calculate the a priori sample size necessary to obtain power (1-ß) > 0.9, effect size = 0.4, and α = 0.05. The result showed a required total sample of 36 subjects. Finally, the sample was established with the maximal volunteer participants in anticipation of possible sample loss.

Fifty-seven professional swimmers completed this study. They were recruited from the Spanish national swimming team. All participants met the inclusion criteria (Table 1) and were informed, in writing and verbally, about the procedures of this study prior to the assessment day.

After receiving detailed information, each participant signed an informed consent form, according with the World Medical Association’s Declaration of Helsinki (2024) [34]. The participant characteristics are shown in Table 2.

### 2.3. Procedures

All participants were recruited from two training camps of the Spanish swimming national team. The swimmers who agreed to participate were interviewed to collect descriptive data (Table 2) and tested for the first time at their training facilities. The participants were randomly divided using the online randomization software Research Randomizer (randomizer.org, accessed on 2 October 2024) into EG1, EG2, and CG (Figure 1). Both EGs were instructed on the technical requirements of the exercises and how to use breathing to count repetitions. They had to develop maximal lumbar extension when inhaling (Figure 2, Exercise 1a) and return to a neutral aligned position while exhaling (Figure 2, Exercise 1b). Moreover, EG1 was instructed on the anatomy and biomechanics of the lumbar spine during sagittal plane movements and were asked to close their eyes and try to focus their attention through imagery on lumbar spine movements while performing the exercises.

Four national coaches were instructed to guide and control the development of the intervention. The intervention protocol involved performing three core exercises in the sagittal plane, synchronized with a controlled breathing pace. Each exercise was executed for a single set of ten breaths at a natural respiratory rate (Figure 1). A 30 s rest period was allocated between exercises to ensure recovery and maintain movement quality. Given the breathing pace of one breath every 5 to 6 s, the total duration of the intervention ranged from 4 to 5 min. To minimize potential nervous system fatigue and maximize neuromuscular activation, the exercises were strategically incorporated as the initial component of the swimmers’ specific warm-up routine.

All subjects were retested after completing 66 sessions over 11 weeks. The same protocol was used at the same time as the first test to avoid differences in the number of sessions completed.

### 2.4. Testing Protocol

An electrogoniometer was used (Transducer TSD130A, Biopac Systems, Inc., Goleta, CA, USA), which was integrated with a computer and Acknowledge 3.0.9 software (Biopac Systems, Inc., Goleta, CA, USA), to assess lumbar electrogoniometer flexion, extension, and total ROM degrees. The equipment was calibrated prior to each testing day to determine the 0° and 90° of each frontal and sagittal plane, but only the sagittal data were analyzed as is the plane of movement of undulatory underwater swimming (UUS). The cranial arm of the goniometer was placed over the D11 and D12 spinal processes, while the lower arm was placed over the sacrum (Figure 3a). Therefore, flexion movements were associated with positive degrees, and extension movements were associated with negative degrees. Lumbar ROM scores were obtained by summing the mean flexion degrees and the mean absolute extension degrees collected in each trial.

The computer was calibrated with a sample rate of 500 Hz. A manual chronometer (Namaste^©^ model 898, Gran Canarias, Spain) was used to identify the interval in seconds over which the subjects maintained each position at the recorded degrees. Different Swiss balls (Gymnic Plus Stability physioballs, TMI, Inc.,Osoppo, Údine, Italy), ranging in diameter from 55 to 90 cm, were used to ensure a correct seated body position, at 90 degrees of the hips and knee flexion, with the feet separated at hip height to increase seated stability (Figure 3b). The ball inflation was checked at 3 bars between tests to ensure that the diameter remained stable. We used three sizes of Swiss balls during the evaluation: 55 cm for subjects between 1.60 and 1.70 m tall, 65 cm for subjects between 1.71 and 1.80 m tall, and 90 cm for subjects between 1.81 m and 1.90 m tall. All tests were completed between 2 and 5 PM by the same primary investigator to minimize fluctuations in circadian lumbar ROM [35].

### 2.5. Statistical Analyses

SPSS Version 28 software (SPSS Inc., Chicago, IL, USA) was used for statistical analysis. The descriptive data of the variables are presented as mean (SD). The distribution of the variables was verified using the Kolmogorov–Smirnov test. A repeated measures MANOVA was conducted to examine the effects of group (EG1, EG2 and CG) on a set of dependent variables (flexion, extension, TROM) measured at two time points (pre- and post-intervention). The analysis aimed to determine if there were significant differences in the multivariate profiles of these variables across groups and over time. Multivariate contrast was performed using univariate contrast to determine differences among dependent variables in each condition. When univariate contrasts showed statistically significant main or interaction effects, pairwise comparisons were carried out using the Bonferroni correction. The level of significance was set at *p* = 0.05. A multivariate contrast was performed using a univariate contrast to determine differences among dependent variables in each condition. The partial eta squared (η^2^p) was used as the effect size of the multivariate and univariate contrasts. When the univariate contrasts showed statistically significant main or interaction effects, pairwise comparisons were carried out using the Bonferroni correction. The level of significance was set at *p* = 0.05. For effect sizes, η^2^p was calculated on the main effects with an interpretation of 0.01, 0.06, and 0.14 as small, medium, and large effect sizes, respectively. Concomitantly, Cohen’s *d* effect size was calculated on all pre- and post-intervention scores, with the interpretation of small (0.2), moderate (0.5), and large (0.8) [36].

## 3. Results

The repeated measures MANOVA showed significant differences in the multivariate profiles across groups and over time. F(_8, 48_) = 3.495, *p* = 0.002. EG1 had non-significant increases in lumbar ROM. EG2 had significant increases in TROM and extension ROM, and CG had no changes. Differences between test and retest values were moderate for flexion scores (F), with noticeable variability across groups. The distributions show distinct group trends, with some groups showing improvement or decline from test to retest (Figure 4).

Regarding the Cohen’s *d* values for flexion in most groups, the Cohen’s *d* values for flexion had a small effect size for all groups, with confidence intervals overlapping zero. However, in EG2, the negative Cohen’s *d* suggests a small decline in scores, but with the confidence interval spanning zero, this effect might not be practically meaningful. For all groups, positive Cohen’s *d* values suggest an improvement in scores on the retest, which are moderate for EG1, large for EG2, and small for CG. TROM had negative Cohen’s *d* values: for EG1, the value was moderate, for CG, the value was small, and there was a large negative effect in EG2 (Table 3).

## 4. Discussion

The aim of this study was to investigate the impact of an 11-week program of repeated maximal lumbar movements with closed eyes and without focused attention on lumbar sagittal ROM in elite swimmers versus executing them solely with the proper exercise technique with controlled breathing. The main finding of this research was that large Cohen’s *d* values for extension and TROM indicate meaningful changes for EG2. These changes may reflect the effectiveness of the intervention. Thus, performing core exercises on the underwater undulatory swimming range of movement, with no extra requirements, but developing the right technique and the complete lumbar extension range of movement, had a high impact on extension but had no meaningful effect on flexion ROM in EG2. A possible explanation for these significant and meaningful differences for EG2 could be in the technical requirements for the exercises without focusing attention on lumbar movement. All of the purposed exercises were developed to achieve maximal individual lumbar extension, and performing maximal flexion was not required. These findings are in line with previous studies that found that dynamic stretches could produce increases in joint range of motion and did not reduce strength [37]. The technical requirement of the purposed exercises asked the participants to develop their maximal active lumbar extension in the initial phase when inhaling and then try to align the lumbar spine in the streamline position when exhaling. This movement had to dynamically stretch the abdominal muscles and produce the contraction of the lumbar erector’s spinae muscles. Nevertheless, there were no stretches on the posterior lumbar muscles, as the required range of movement stopped on the neutral aligned position.

However, the small effect sizes suggest stable performance across tests and retests for all measures in EG1. This could indicate that the intervention for EG1 did not strongly influence variability in the EG1-specific intervention and had little immediate impact on flexibility or range of motion over short-term measurements. The specific intervention for EG1 involved performing the exercises with closed eyes while focusing attention on lumbar movements, following the same technical requirements as EG2. The results of this study showed that introducing conscious control and focusing attention on a movement could increase proprioceptive spindle activity, as proprioception is increased by focusing attention and avoiding visual input [32,38]. Increased muscle spindle activity could assist in reducing the maximal stretch of the abdominal muscles and, therefore, reduce the lumbar extension limit on EG1 on each repletion [38,39]. Thus, by reducing the abdominal length, the proprioception system seems to enhance the action to create adjusted abdominal contraction for the lumbar alignment after maximal controlled extension in each repetition for each exercise in EG1. Moreover, EG1 was instructed on lumbar anatomy to be able to practice on imagery of the lumbar spine movements and the surrounding muscle activity. Imagery in sport is often used to prepare athletes for competition [40]. Motor imagery is based on the activity of specific neural networks. Researchers have elucidated many of these areas and pathways by investigating brain activity during motor imagery. Brain activation during the imagery of an action is stronger when sensory inputs are like those that occur during the real execution of the same action [41]. In the present study, we applied specific cognitive imagery at the time that participants performed the exercises by focusing on the lumbar spine movements during the execution. Furthermore, EG1 was instructed to improve individual differences in the ability to create vivid motor imagery by providing lumbar anatomy and biomechanics lessons prior to starting the intervention. Focused attention on movement execution has been shown to influence neuromuscular activation by enhancing motor unit recruitment and proprioceptive sensitivity. In our study, the EG1 group was instructed to direct their attention toward lumbar movement while performing the exercises, which may have influenced muscle spindle sensitivity and overall motor control. This instruction could lead to the major activation of neurophysiological motor control actions during the intervention and assist in reducing the possibility of stretching the abdominal muscles too much, allowing the participants to perform more controlled and intense contractions during lumbar flexion movements in all three exercises and thus reducing their length and total lumbar extension after 11 weeks.

A short core workout was included at the beginning of the warm-up in the morning session. The purpose was to not disrupt the habitual practice too much and also to perform the exercises in a condition where the nervous system is not fatigated and ready to focus on a technical action, as it is reported that central nervous system fatigue can negatively impact motor control and proprioception [42,43,44,45,46]. Based on the ecological validity of the intervention, and considering that the participants should complete 11 weeks of six sessions per week, we decided not to include more than three exercises and just one set of 10 repetitions. In this way, we tried to control the adherence to the intervention and the minimal loss to follow-up. Despite this, we lost 10 participants in the CG and 11 in the EG. In an elite environment, it is important not only to respect the ecological validity of the studies but also to design integrated workouts to allow athletes, coaches, and strength and conditioning professionals to achieve performance goals with a minimum loss of time.

It is accepted that core workouts should include exercises that involve the same muscle chains of the specific sport technique to obtain high transference to performance requirements [47,48,49]. Nevertheless, there is still a lack of research on specific core workouts for improving specific swimming techniques, as most of the reported programs use the conventional core exercises, which include basic core movements that could be part of any training program for any athlete, without specifying the sport [9,10,11,12,48]. For example, in swimming, the effects of 6-week core exercises on the swimming performance of national level swimmers were investigated. The program included Flutter kicks (scissors), Single-leg V-ups, Prone physio ball trunk extension, and Russian twists. This non-functional core training program caused non-significant improvement in the number of swimming variables, which together resulted in an overall increase in 50 m front crawl swimming performance by 1.2% in the EG, whereas the CG swimmers improved their performance by just 0.7% [12]. Another research study aimed to investigate the effect of a 12-week dry-land core training program on physical fitness and swimming performance in elite adolescent swimmers. The core training program consisted of 4 weeks of core stabilization (bridge, plank to push-up, and bird dog), 4 weeks of core muscular power (deadlifts, squats, and rows were conducted using a single arm or leg for resistance exercise), and 4 weeks of power endurance (core training motions, such as medicine ball slams, one-arm dumbbell snatch, and chop exercises were conducted). The 12-week dry-land core training program resulted in statistically significant improvements in anaerobic power, core stability, upper extremity muscular endurance, and swimming performance, although that research study used non-specific core exercises [11]. Additionally, 6 weeks of three sessions per week of a non-functional core-training program, along with regular swimming training, significantly improved the freestyle swimming performance and core muscle properties, such as contractility, excitability, extensibility, and elasticity, of the young swimmers experimental group compared with the other group. However, the authors stated that these results can only be applied to the same age group, as the participants included 18 young recreational swimmers (13 ± 2 years) with different maturative ages [10].

The present study was designed for specific lumbar gestures on underwater undulatory swimming. It aimed to reproduce the synergies of three moments of UUS: those involved in the transition from the upward kick to the downward kick, the downward kick, and the upward kick. The participant distribution in our study was as follows: 31.5% specialized in freestyle, 24.5% in breaststroke, 15.7% in butterfly, 14% in backstroke, and 14% in individual medley (Table 2). Each swimming stroke places different biomechanical demands on the body, particularly on the spine and core musculature. Given that UUS plays a crucial role in starts and turns across all strokes—especially in backstroke and butterfly—variations in stroke specialization could have influenced the adaptations observed in lumbar ROM. Swimmers specializing in butterfly and backstroke rely more heavily on UUS for propulsion, requiring a greater degree of lumbar flexibility and control to optimize wave amplitude and frequency [15]. Conversely, freestyle and breaststroke swimmers may not depend as much on UUS but still benefit from improved lumbar mobility in starts, turns, and transitions. Individual medley swimmers, who must adapt their movement patterns across all strokes, likely require a balanced combination of flexibility and neuromuscular coordination [14,15]. Recent studies have highlighted the importance of lumbar range of motion (ROM) in enhancing undulatory underwater swimming (UUS) performance among elite athletes. A systematic review by Veiga et al. [15] emphasized that a greater ROM of the lower trunk during body undulations is associated with faster forward velocities during UUS. This increased ROM allows swimmers to achieve more effective undulatory movements, thereby improving propulsion and overall performance. Moreover, we conducted a post hoc analysis to assess whether the loss of participants altered the initial distribution of swimming specialties between the control group (CG) and the experimental group (EG). Our analysis confirmed that the distribution of swimming styles remained proportionally balanced in both groups after participant loss, ensuring that no single stroke category was disproportionately affected. Since the primary objective of this study was to assess changes in lumbar range of motion (ROM) rather than stroke-specific adaptations, maintaining an even distribution of swimmer specialties across groups helped minimize potential bias in the results.

All three exercises required the EG to reach maximal lumbar extension while inhaling, activating the erector spinae (ES) and multifidus (MF), and to reach for the streamline lumbar position when exhaling, activating the rectus abdominis (RA), internal abdominal muscle (IO), and the transversus abdominis (TrA) at prone, four tab, and supine lying positions. Exercises 1 and 2 were more demanding for ES and MF, as gravity acted against the anterior pelvic tilt and lumbar extension movements. Exercise 3 was more demanding for RA and IO, as gravity interferes with developing posterior pelvic tilt and lumbar flexion. In UUS in elite swimmers, both the upward kick and downward kick follow the trunk muscles involved in the pelvic forward–backward tilt movement [13,14]. Moreover, flexor and extensor muscles in the trunk, thigh, and leg are required to contract alternately during dolphin kicking [13,14,15,16]. Functional movement patterns play a very important role in achieving high levels of sports performance. It is accepted that it is important to define athletes’ functional movement in their swimming techniques and to follow joint stability and mobility regularly as the functional movement patterns of an athlete that are not proper have negative effects on athlete performance [48].

Although it was not an aim of this study, there was an intentionality in our synchronization of breath. We asked the participants to inhale with the lumbar extension and exhale to reach the lumbar streamline position. We proposed this breathing pattern to increase the demands of the involved core muscles, as it is well studied that inspiratory muscles involve some ES and MF, while the main voluntary expiratory muscles are the TrA, RA, IO, and major and minor obliques [50,51,52]. Moreover, several studies agreed that controlling the breathing pattern during exercise can improve pelvic posture and lumbar alignment [53,54]. By adding voluntary expiration with lumbar flexion, we also looked for additional activation of the TrA. The TrA has been demonstrated to be one of the most important muscles in increasing core stability and stiffness; therefore, it is important in reducing drag on UUS [10,11,49,55,56].

Regarding the testing procedures and instruments, we decided to assess lumbar ROM with an electrogoniometer in a seated position on a Swiss ball, as this seated position allowed for free lumbar movements in the sagittal plane. The stable seated position involved having both feet on the floor, separated at hip height, as this position allowed the swimmers to produce their maximal flexion and extension movements without discomfort, limitation, or disbalance. The most reliable and valid instruments to assess spine ROM are X-rays and MRI, but these instruments are associated with radiation and can be expensive [3,57]. Perriman et al. [58] measured subjects in static positions using an electrogoniometer because the “clinical gold standard” is a static measurement [58]. We decided to assess active lumbar movements as there is evidence of better reliability for the active movements test than the passive movements test. In addition, only an experienced researcher conducted all the measurements, as poor reliability is often caused by inexperienced researchers and difficulties placing the device sensors in the right places [59,60].

One of the primary limitations of this study was the loss of participants during the retest phase, which could have influenced the robustness of our findings. Despite initially enrolling enough participants, 10 swimmers from the control group (CG) and 11 from the experimental group (EG) did not complete the intervention due to factors such as training schedules, competition commitments, and unforeseen circumstances. This loss of participants may have reduced the statistical power of this study, potentially affecting the generalizability of the results. While post hoc analyses confirmed that the distribution of swimmer specialties remained balanced after these losses, ensuring no significant bias in stroke specialization, the decrease in sample size may still limit the ability to fully capture inter-individual variability in response to the intervention. Future studies should consider implementing strategies to improve participant retention, such as shorter intervention durations, individualized follow-up protocols, or structured incentives to encourage continued participation.

Despite these limitations, the significant findings observed in lumbar range of motion (ROM) improvements highlight the practical applications of this core training program. However, caution should be exercised when extrapolating these results to larger or more diverse swimmer populations, and replication studies with a higher participant retention rate are warranted to confirm these findings.

One limitation of this study is that it focused primarily on the effects of lumbar range of motion (ROM) and the biomechanical aspects of breathing without assessing pulmonary function or other factors related to UUS performance. Additionally, participant loss in both the experimental and control groups posed a challenge.

Future research should consider evaluating the influence of specific short-duration core training on the ability to reproduce lumbar alignment, as well as the effects of this 11-week intervention on UUS time and depth. A more detailed analysis is also warranted to explore the outcomes of performing exercises with closed eyes, focused attention, and imagery of lumbar movement, as implemented in EG1. This aspect of the intervention did not result in significant improvements in lumbar ROM, unlike EG2. Further investigations are needed to evaluate the role of attentional focus and visual suppression in neuromuscular engagement. Additionally, in our future work, we will emphasize the practical applications of our findings, particularly in training and rehabilitation settings, where optimizing motor control and proprioception may enhance performance and injury prevention. We will also determine the effects of these three core exercises on proprioceptive outcomes and their potential role in enhancing UUS performance, particularly in optimizing gliding body position and movement efficiency.

## 5. Conclusions

Applying an 11-week program of three core exercises performed six times per week at a controlled breathing pace—coordinating inspiration with maximal extension and expiration with the lumbar streamline position—led to a significant and meaningful increase in lumbar extension and total range of motion (TROM) in 20 elite swimmers. However, performing the same exercises with closed eyes, focusing attention on the lumbar spine, and using imagery of lumbar movements did not result in a significant improvement in lumbar movement outcomes in the sagittal plane. Coaches and strength and conditioning professionals can incorporate this 11-week core training program into swimming training regimens to enhance lumbar extension and total range of motion (TROM). The exercises should be performed six times per week, synchronized with breathing, to optimize spinal mobility without adding excessive fatigue. Moreover, since lumbar extension plays a key role in generating propulsion during UUS, implementing this program may help swimmers improve the coordination of impulse generation, leading to more effective underwater movement, particularly for butterfly, backstroke, and starts/turns in all strokes. Other practical applications for injury prevention and rehabilitation could be that improved lumbar mobility and flexibility may contribute to reducing the risk of lower back stiffness or imbalances caused by repetitive swimming motions. This program could be beneficial not only for performance enhancement but also as a preventive or rehabilitative measure for swimmers prone to lower back issues. While this study demonstrated significant improvements in lumbar extension with the breathing-coordinated exercises, practitioners should consider individualizing training intensity and volume based on the athlete’s needs, ensuring that lumbar mobility gains translate effectively into stroke efficiency and overall swimming performance. Finally, this program provides a practical, time-efficient approach to improving lumbar flexibility and mobility in elite swimmers, with direct implications for performance enhancement, injury prevention, and training methodology.

## Figures and Tables

**Figure 1 healthcare-13-00457-f001:**
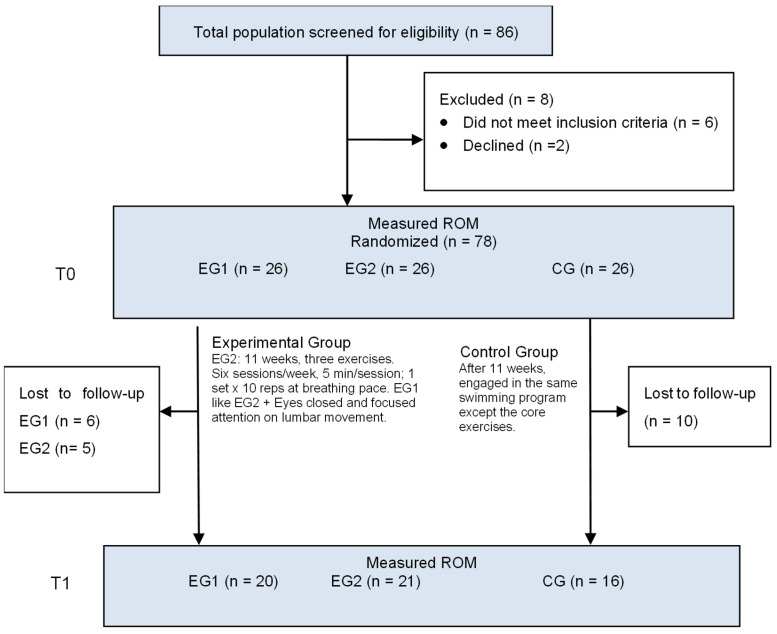
Flowchart. T0 = baseline test, T1 = retest. EG1 = experimental group 1; EG2 = experimental group 2; CG = control group; ROM = range of motion.

**Figure 2 healthcare-13-00457-f002:**
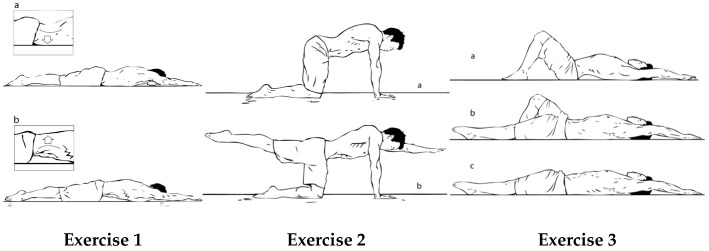
Exercise 1a shows the details of the required extension while inhaling. Exercise 1b shows the details of lumbar alignment while exhaling. Exercise 2a shows the details of the required extension while inhaling. Exercise 2b shows the details of lumbar and body alignment while exhaling. Exercise 3a shows the details of the required lumbar extension while inhaling. Exercise 3b shows the details of lumbar and body alignment while exhaling at first level of difficulty. Exercise 3c shows the details of lumbar and body alignment while exhaling at second level of difficulty.

**Figure 3 healthcare-13-00457-f003:**
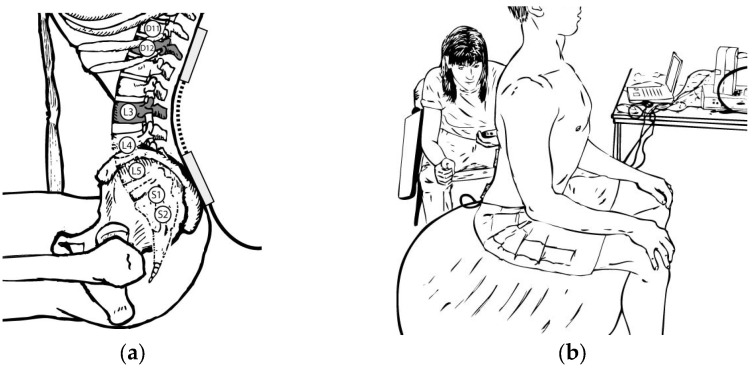
Testing procedures: (**a**) electrogoniometer placement (D11 = 11th dorsal vertebra, D12 = 12th dorsal vertebra, L3 = 3rd lumbar vertebra, L4 = 4th lumbar vertebra, L5 = 5th lumbar vertebra, S1 = 1st sacral vertebra, S2 = 2nd sacral vertebra) (**b**) testing position.

**Figure 4 healthcare-13-00457-f004:**
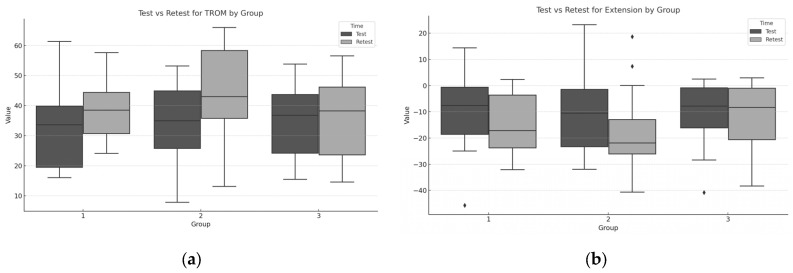
Boxplots for each dependent variable expressed by degrees, comparing test and retest values across the three groups: (**a**) boxplot for TROM values; (**b**) boxplot for lumbar extension variable.

**Table 1 healthcare-13-00457-t001:** Inclusion criteria.

Part of the national Spanish swimming team. Minimum 4 years of experience at this high level. Did not suffer from any ailment or discomfort that would prevent him/her from competing, performing the exercises, or lumbar range of motion. Achieved an elite status and held international rankings in their respective age categories. Did not take medications throughout this study. Free of musculoskeletal injuries during the previous three months.

**Table 2 healthcare-13-00457-t002:** Descriptive characteristics of the participants.

Age (yr), mean (SD)	
Males	20.2 (4.2)
Females	20.7 (3.3)
Gender, n female (%)	23 (40.3%)
Body mass Kg, mean (SD)	69.7 (10.3)
Height, cm, mean (SD)	177.8 (7.5)
Professional swimming experience (years) mean (SD)	8.7 (4.4)
Level, n (%)	
Olympic	33 (57.8%)
International	13 (22.8%)
National	11 (19.2%)
Main swimming style n (%)	
Freestyle	18 (31.5%)
Breaststroke	14 (24.5%)
Butterfly	9 (15.7%)
Backstroke	8 (14.0%)
Individual medley	8 (14.0%)
Main competition distance n (%)	
50–100	20 (35.1%)
200–400	31 (54.4%)
800–1500	6 (10.5%)

**Table 3 healthcare-13-00457-t003:** Test and retest descriptive values and Cohen’s *d* effect size for flexion (Flex), extension (Ext), and TROM.

Group	Flex Test	Flex Retest	*d*(CI)	Ext Test	Ext Retest	*d*(CI)	TROM Test	TROM Retest	*d*(CI)
EG1	22.8	23.5	−0.10	−9.6	−14.5	0.47	32.4	38.0	0.52
	(8.6)	(5.6)	(−0.48)	(14.2)	(10.5)	(−0.15)	(12.9)	(9.4)	(−0.85)
EG2	24.1	26.4	−0.47	−10.2	−18.3 *	0.83	34.4	44.7 *	−1.12
	(6.4)	(6.1)	(−0.78)	(14.2)	(15.1)	(−0.05)	(12.0)	(14.0)	(−1.11)
CG	23.5	24.0	−0.38	−11.2	−11.9	0.16	34.7	35.9	−0.31
	(7.8)	(7.6)	(−0.52)	(11.9)	(12.4)	(−0.45)	(11.9)	(12.5)	(−0.64)

Note: Results are expressed by degrees, mean (SD). Significant differences are marked with bolt and * (*p* < 0.05). EG1 = experimental group 1, EG2 = experimental group 2, CG = control group. Test = lumbar baseline test. Retest = lumbar degrees after 11 weeks.

## Data Availability

Data is unavailable due to privacy and ethical restrictions.

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
