# Peer review of "Effect of an 11-Week Repeated Maximal Lumbar Movement with Controlled Breathing on Lumbar Sagittal Range of Motion in Elite Swimmers: A Randomized Clinical Trial"

_healthcare, 2025, doi:10.3390/healthcare13050457_

Round 1

Reviewer 1 Report

Comments and Suggestions for Authors

This title is not grammatically correct

Line 14- *in that technique

Line 16- "an 11-week"?, for underwater technique on ROM?

21- three sets of ten

The results in the abstract need clarified- the p values are stated and then non statistical differences are stated. This title also does not seem to match the difference between EG1 and EG2

Ln41-44 need rewritten for clarity

53- depends

66-  grammar

68- which study?

73- fix for clarity

78-80 grammar.

85- this purpose does still not make sense, nor does it account for the differences described in the experimental groups in the abstract

the flowchart and your methods should depict the differences between EG1 and EG2. It is unclear what the difference is, or why the difference was created, as it was not mentioned at all in the intro. 

;line 132-135 need to be rewritten for clarity 

211- again, reworded for clarity

235- than the EG2?

317- it had to have been an aim, as that is how you separated your experimental groups?

346- needs rewritten

this article has a lot of grammatical errors that make it difficult to follow. your experimental groups are also not well explained, and the title does not match the difference in your two groups. Your intro needs to talk about these differences in purposeful breathing etc., between EG1 and EG2 to allow some better understanding of what occurred. The discussion seems a little jumpy to this point as well, as it seems like an afterthought, but your entire study design was based on this.

Comments on the Quality of English Language

there are a lot of English and grammar issues that are impairing understanding of the content.

Author Response

Generic comments

Authors: We would like to thank again the Reviewer 1 for his/her time for making constructive feedback to improve our manuscript.

Comment 1 This title is not grammatically correct

Response 1: Thank you very much for your comment. We agree with this comment. We have amended as follows:

[Effect of an 11-Week Repeated Maximal Lumbar Movement with Controlled Breathing on Lumbar Sagittal Range of Motion: A Randomized Clinical Trial]

Comment 2: Line 14- *in that technique

Response 2: Thank you very much for your comment. We agree with this comment. We have amended.

Comment 3: Line 16- "an 11-week"?, for underwater technique on ROM?.

Response 3: Thank you very much for your comment. We agree with this comment. We have amended as follows: [The aim of this study was to investigate the impact of an 11-week program of repeated maximal lumbar movements with closed eyes and without focused attention versus executing them solely with proper exercise technique with controlled breathing on lumbar sagittal ROM in elite swimmers]

Comment 4: 21- three sets of ten

Response 4: Thank you, amended

Comment 5:

The results in the abstract need clarified- the p values are stated, and then non statistical differences are stated. This title also does not seem to match the difference between EG1 and EG2

Response 5: Thank you very much for your comment. We agree with this comment. We have amended as follows: [EG2 only followed the technique of the exercises at a controlled breathing pace.] We also changed the title to match the conclusions as follows: [Effect of an 11-Week Repeated Maximal Lumbar Movement with Controlled Breathing on Lumbar Sagittal Range of Motion in elite swimmers: A Randomized Clinical Trial].

Comment 6: Ln41-44 need rewritten for clarity

Response 6: Thank you. We have amended as follows: […Normal lumbar range of motion (ROM) enables the body to better respond to external forces and helps prevent most lumbar injuries and pain. A significant reduction in lumbar ROM, particularly in the sagittal plane, has been observed in individuals with low back pain. Moreover, those with low back pain tend to experience a general decrease in lumbar ROM].

Comment 7: 53- depends

Response 7: Thank you. Amended.

Comment 8: 66- grammar

Comment 9: 68- which study?

Response 8 and 9: Thank you for your comment. Amended as follows: [More specifically, measurements were taken for the following core muscles of each swimmer: rectus abdominis (RA), internal oblique (IO), erector spinae (ES), and multifidus (MF). The study of Matsuura et al. (2020) analyzed muscular synergies at three moments of the UUS: the transition from the upward kick to the downward kick, the downward kick, and the upward kick. In the UUS of elite swimmers, both the upward and downward kicks followed the activation pattern of trunk muscles involved in the pelvic forward-backward tilt movement, along with the activation of muscles in the lower limbs].

Comment 10: 73- fix for clarity

Response 10: Thank you. Amended as follows: [Core exercises are commonly designed to enhance core motor control or strength in athletes and individuals with low back pain].

Comment 11: 78-80 grammar.

Response 11: Thank you very much! Amended as follows: [Core workouts are essential components of swimming strength and conditioning programs and are often incorporated into dryland warm-ups in swimming practice. The significant improvement in core muscular function contributes to enhanced swimming performance].

Comment 12: 85- this purpose does still not make sense, nor does it account for the differences described in the experimental groups in the abstract

Response 12: Thank you very much. Amended as follows: [Therefore, this study aimed to investigate the impact of an 11-week program of repeated maximal lumbar movements with closed eyes and without focused attention versus executing them solely with proper exercise technique with controlled breathing on lumbar sagittal range of motion in elite swimmers. We hypothesized that those training programs would lead to a significant increase in swimmers' total lumbar range of motion compared to the control group.]

Comment 13: the flowchart and your methods should depict the differences between EG1 and EG2. It is unclear what the difference is, or why the difference was created, as it was not mentioned at all in the intro. 

Response 13: Thank you. We add extra information in the introduction section, as in the methods section, too.

Comment 14: line 132-135 need to be rewritten for clarity.

Response 14: Thank you. Amended as follows: [The intervention protocol involved performing three core exercises in the sagittal plane, synchronized with a controlled breathing pace. Each exercise was executed for a single set of ten breaths at a natural respiratory rate (Figure 1). A 30-second rest period was allocated between exercises to ensure recovery and maintain movement quality. Given the breathing pace of one breath every 5 to 6 seconds, the total duration of the intervention ranged from 4 to 5 minutes. To minimize potential nervous system fatigue and maximize neuromuscular activation, the exercises were strategically incorporated as the initial component of the swimmers' specific warm-up routine.]

Comment 15: 211- again, reworded for clarity

Response 15: Thank you. Amended as follows: [The aim of this study was to investigate the impact of an 11-week program of repeated maximal lumbar movements with closed eyes and without focused attention versus executing them solely with proper exercise technique with controlled breathing on lumbar sagittal ROM in elite swimmers]

Comment 16: 235- than the EG2?

Response 16: Thank you. We amended as follows: [The specific intervention for EG1 involved performing the exercises with closed eyes while focusing their attention on lumbar movements, following the same technical requirements as EG2]

Comment 17: 317- it had to have been an aim, as that is how you separated your experimental groups?

Response 17: Thank you. We respectfully disagree, as proper breathing is a fundamental component of all core exercise programs. In fact, both experimental groups followed the same breathing pattern. The only difference between EG1 and EG2 has been explained in previous responses.

Comment 18: 346- needs rewritten

Response 18: Thank you, amended as follows: [One limitation of this study is that it focused primarily on the effects of lumbar range of motion (ROM) and the biomechanical aspects of breathing, without assessing pulmonary function or other factors related to UUS performance. Additionally, participant attrition in both the experimental and control groups posed a challenge. Future research should consider evaluating the influence of specific short-duration core training on the ability to reproduce lumbar alignment, as well as the effects of this 11-week intervention on UUS time and depth. A more detailed analysis is also warranted to explore the outcomes of performing exercises with closed eyes, focused attention, and imagery of lumbar movement, as implemented in EG1. This aspect of the intervention did not result in significant improvements in lumbar ROM, unlike EG2.Further investigations are needed to determine the effects of these three core exercises on proprioceptive outcomes and their potential role in enhancing UUS performance, particularly in optimizing gliding body position and movement efficiency.]

Comment 19: this article has a lot of grammatical errors that make it difficult to follow. your experimental groups are also not well explained, and the title does not match the difference in your two groups. Your intro needs to talk about these differences in purposeful breathing etc., between EG1 and EG2 to allow some better understanding of what occurred. The discussion seems a little jumpy to this point as well, as it seems like an afterthought, but your entire study design was based on this.

Response 19: Thank you for your detailed feedback. We appreciate your time in reviewing our work and providing constructive criticism. We acknowledge the concerns regarding grammatical clarity and will carefully revise the manuscript to improve readability and coherence. Additionally, we will ensure that the description of the experimental groups is clearer, providing a more precise explanation of their differences. Regarding the title, we recognize the importance of aligning it more accurately with the distinctions between EG1 and EG2, and we will consider refining it to better reflect the study design. We also take note of your suggestion to elaborate on purposeful breathing in the introduction to provide a clearer context for the reader. Lastly, we will refine the discussion section to improve its flow and ensure it aligns more cohesively with the study’s main objectives. Your feedback is invaluable, and we truly appreciate your insights in helping us strengthen our work.

Reviewer 2 Report

Comments and Suggestions for Authors

Optimizing muscle extensibility and joint range of motion is one of the objectives of many exercise programs. Many authors consider strengthening to be as effective as stretching in improving range of motion. Changes in range of motion involve physiological, structural and neurological adaptations.

Progressive strengthening combined with a gradual increase in range of motion may lead to functional improvement more rapidly than isolated stretching.

The axial musculature contains numerous extrapyramidal type I fibers. The choice of respiratory control favoring aerobic metabolism and synergy with amplitude in the sagittal functional plane is a judicious one. 

1. The aim seems to be twofold. The gain in amplitude must be complemented by the cognitive neurological aspect. Two groups, EG1 and EG2, each with its own objective.

2. The introduction should address the neurological aspect for the EG1 group, with suppression of visual afferents and axial proprioceptive concentration.

Is the objective of eye closure to improve balance?

Why combine both (eyes & concentration)?

Are mirror neurons involved? 

3. Improvement in extension is well explained by the choice of exercises, but concentration, which in principle improves muscular activity, needs to be better discussed.

4. The conclusion should include both objectives and their practical implications.

Author Response

Comment 1:

Optimizing muscle extensibility and joint range of motion is one of the objectives of many exercise programs. Many authors consider strengthening to be as effective as stretching in improving range of motion. Changes in range of motion involve physiological, structural and neurological adaptations.

Progressive strengthening combined with a gradual increase in range of motion may lead to functional improvement more rapidly than isolated stretching.

The axial musculature contains numerous extrapyramidal type I fibers. The choice of respiratory control favoring aerobic metabolism and synergy with amplitude in the sagittal functional plane is a judicious one. 

Response 1: Thank you for your insightful comment. We appreciate your perspective on muscle extensibility, joint range of motion, and the role of strengthening in improving flexibility. Indeed, recent research highlights that progressive strengthening, when combined with a gradual increase in range of motion, can lead to more functional improvements compared to isolated stretching. We also acknowledge the importance of axial musculature, particularly its high proportion of type I fibers, in maintaining postural control and movement efficiency. The choice to incorporate respiratory control in our intervention aimed to enhance neuromuscular coordination and synergy, particularly within the sagittal functional plane, which aligns with the principles of efficient motor control and movement economy.Your feedback reinforces the rationale behind our study design, and we will consider incorporating additional discussion on these points to further clarify the physiological, structural, and neurological adaptations associated with our intervention.

Comment 2: The aim seems to be twofold. The gain in amplitude must be complemented by the cognitive neurological aspect. Two groups, EG1 and EG2, each with its own objective.

Response: Thank you for your thoughtful observation. Indeed, our study aims to address both the gain in lumbar range of motion (ROM) and the cognitive-neurological aspects of movement control. The intervention was designed to explore whether different sensory and attentional conditions influence motor outcomes, with EG1 focusing on closed eyes and attentional engagement, while EG2 followed the same technical requirements without these additional cognitive demands. By distinguishing these two experimental approaches, we sought to assess whether the incorporation of cognitive and proprioceptive elements provides an advantage over traditional execution in improving lumbar ROM. Your comment aligns with the rationale behind our study design, and we appreciate your perspective on the complementary relationship between amplitude gain and neurological engagement. In fact, we are working on a second article to further explain the neurological implications of EG1.

Comment 3:

  1. The introduction should address the neurological aspect for the EG1 group, with suppression of visual afferents and axial proprioceptive concentration.

Is the objective of eye closure to improve balance?

Why combine both (eyes & concentration)?

Are mirror neurons involved? 

Response 3: Thank you for your insightful comments and questions. We appreciate your perspective on the neurological aspects of the EG1 group and the role of visual suppression and proprioceptive focus in our intervention.We expanded the introduction to better explain the neurological rationale behind EG1. The suppression of visual afferents aims to shift reliance toward proprioceptive feedback, particularly from axial musculature and muscle spindles, which play a key role in motor control and postural regulation.

Regarding the objective of eye closure, while balance improvement is a potential secondary effect, our primary goal was to enhance proprioceptive awareness and neuromuscular engagement during lumbar movements. By reducing visual input, participants were encouraged to rely more on internal sensory mechanisms, which may contribute to improved motor control and body awareness.

The combination of eye closure and attentional focus was implemented to investigate whether conscious engagement with lumbar movement could further enhance neuromuscular activation. This approach aligns with previous findings on motor learning, where attentional strategies can influence movement efficiency and coordination.

As for mirror neuron involvement, while our study did not explicitly measure their activation, the principles of motor imagery and focused attention suggest that neural circuits related to movement observation and execution could be engaged. Future research could explore this connection further, as it may provide deeper insights into the cognitive-motor interactions underlying proprioceptive training.

We sincerely appreciate your thought-provoking questions and will incorporate these considerations into the manuscript to clarify our study design and its underlying mechanisms.

Comment 4: Improvement in extension is well explained by the choice of exercises, but concentration, which in principle improves muscular activity, needs to be better discussed.

Response 4: Thank you for your valuable feedback. We appreciate your acknowledgment of the rationale behind the exercise selection for improving extension.

Regarding the role of concentration in enhancing muscular activity, we recognize the need for a more in-depth discussion. Focused attention on movement execution has been shown to influence neuromuscular activation by enhancing motor unit recruitment and proprioceptive sensitivity. In our study, the EG1 group was instructed to direct their attention toward lumbar movement while performing the exercises, which may have influenced muscle spindle sensitivity and overall motor control.

We will expand our discussion to better address how attentional focus interacts with proprioception and neuromuscular engagement, drawing from existing literature on motor learning and sensorimotor control. Your suggestion will help us clarify the potential impact of concentration on performance outcomes.

Comment 5: The conclusion should include both objectives and their practical implications.

Response 5: Thank you for your valuable feedback. We appreciate your suggestion Your insight helps us improve the clarity and applicability of our conclusions, and we sincerely appreciate your contribution. We amended as follows in conclusion section: [Applying an 11-week program of three core exercises, performed six times per week at a controlled breathing pace—coordinating inspiration with maximal extension and expiration with the lumbar streamline position—led to a significant and meaningful increase in lumbar extension and total range of motion (TROM) in 20 elite swimmers. However, performing the same exercises with closed eyes, focusing attention on the lumbar spine, and using imagery of lumbar movements did not result in a significant improvement in lumbar movement outcomes in the sagittal plane. Coaches and strength and conditioning professionals can incorporate this 11-week core training program into swimming training regimens to enhance lumbar extension and total range of motion (TROM). The exercises should be performed six times per week, synchronized with breathing, to optimize spinal mobility without adding excessive fatigue. Moreover, since lumbar extension plays a key role in generating propulsion during UUS, implementing this program may help swimmers improve the coordination of impulse generation, leading to more effective underwater movement, particularly for butterfly, backstroke, and starts/turns in all strokes. Other practical application for injury prevention and rehabilitation could be that improved lumbar mobility, and flexibility may contribute to reducing the risk of lower back stiffness or imbalances caused by repetitive swimming motions. This program could be beneficial not only for performance enhancement but also as a preventive or rehabilitative measure for swimmers prone to lower back issues. While the study demonstrated significant improvements in lumbar extension with the breathing-coordinated exercises, practitioners should consider individualizing training intensity and volume based on the athlete's needs, ensuring that lumbar mobility gains translate effectively into stroke efficiency and overall swimming performance. Finally, this program provides a practical, time-efficient approach to improving lumbar flexibility and mobility in elite swimmers, with direct implications for performance enhancement, injury prevention, and training methodology].

Reviewer 3 Report

Comments and Suggestions for Authors

Dear authors,

I would like to congratulate you on the scientific rigor of your study, which proposes a relevant methodological approach and provides interesting insights into the impact of short functional training on lumbar sagittal amplitude in elite swimmers. Your work represents an interesting contribution to our understanding of muscular (biomechanical) adaptations in this context.

In order to refine certain points and enrich the discussion, I'd like to submit a few comments and questions to you:

- It might be relevant to specify how the population of swimmers is distributed according to their swimming specialty. Indeed, each type of stroke (front crawl, back crawl, butterfly, breaststroke) places different demands on muscles and joints, particularly in certain areas of the spine. This distinction is essential, as it could influence physiological/muscular adaptations.

- You mention an attrition of 10 participants in the control group (CG) and 11 in the experimental group (EG), which is understandable in the context of a long-term follow-up protocol. However, has this loss altered the initial balance of groups according to swimmers' specialities? A subsequent analysis of this distribution would enable us to assess whether it may have introduced a bias in the results obtained.

- On a broader level, have you considered the existence of a group constitution effect that might influence the results? A comparative analysis of inter-group differences at the start and end of the study could help refine the interpretation of the data and anticipate possible confounding factors.

- Although the proposed exercises were carried out over a period of 11 weeks at a rate of 6 sessions per week, their impact could be moderate in comparison with training loads specific to swimming. This discipline involves particular techniques and targeted muscular demands which could modulate the effects of the complementary exercises. Have you taken this interaction into account in your analysis of the results?

- Why did you choose a ball rather than a rigid support that would allow more precise adaptation to the participants' morphological characteristics (e.g.: precise adjustment of the knee angle to 90°)? This parameter could have an impact on the reproducibility and standardization of the exercises, and might be worth discussing (even if you're talking about active gainage/attitude).

- Have you considered the influence of breathing tempo on exercise execution? Depending on the breathing rate adopted by participants, isometric maintenance could be induced, thus influencing muscle recruitment and potentially the results observed.

- The combined use of a goniometer and an electromyographic acquisition system could have enabled a more detailed analysis of muscle recruitment and the joint amplitudes involved. Such an approach could have strengthened the validity and accuracy of the conclusions.

- A more detailed discussion of muscle activation during the exercises would have been interesting. I recognize the difficulty of precisely characterizing the involvement of deep muscles, but an analysis of superficial muscles could have been envisaged. This would have enriched the interpretation of the results and enabled them to be compared with those of previous studies.

Your study represents a significant contribution to the analysis of functional training applied to elite swimmers, which should be pursued, as I believe that the effectiveness of execution in mental representation / eyes closed had proved its effectiveness in other sports, particularly in bodybuilding. These remarks are simply intended to question the methodological and analytical robustness of your work.

Yours sincerely

Author Response

Comment 1:

Dear authors,

I would like to congratulate you on the scientific rigor of your study, which proposes a relevant methodological approach and provides interesting insights into the impact of short functional training on lumbar sagittal amplitude in elite swimmers. Your work represents an interesting contribution to our understanding of muscular (biomechanical) adaptations in this context.

Response 1: Thank you very much for your kind words and thoughtful feedback. We sincerely appreciate your recognition of the scientific rigor and methodological approach of our study. It is gratifying to know that our research provides valuable insights into the biomechanical adaptations associated with short functional training in elite swimmers. Your encouraging comments motivate us to continue refining our work and exploring further applications of our findings. We are grateful for the opportunity to contribute to the understanding of neuromuscular and biomechanical adaptations in this context.

Comment 2:

It might be relevant to specify how the population of swimmers is distributed according to their swimming specialty. Indeed, each type of stroke (front crawl, back crawl, butterfly, breaststroke) places different demands on muscles and joints, particularly in certain areas of the spine. This distinction is essential, as it could influence physiological/muscular adaptations.

Response 2:

Thank you for your insightful comment. We appreciate your suggestion to specify the distribution of swimmers according to their swimming specialty.

Indeed, different strokes impose distinct biomechanical demands on muscles and joints, particularly affecting spinal movement and muscular adaptations. This differentiation is crucial for understanding potential variations in physiological responses to the intervention. While our study primarily focused on the general effects of core training on lumbar range of motion, we recognize that stroke-specific demands may influence outcomes.

We have clarified the distribution of participants based on their primary swimming specialty and discuss how this factor might impact the interpretation of our results. Your suggestion adds valuable depth to our analysis, and we sincerely appreciate your contribution.

Comment 3:

You mention an attrition of 10 participants in the control group (CG) and 11 in the experimental group (EG), which is understandable in the context of a long-term follow-up protocol. However, has this loss altered the initial balance of groups according to swimmers' specialities? A subsequent analysis of this distribution would enable us to assess whether it may have introduced a bias in the results obtained.

Response 3: Thank you for your thoughtful comment and for highlighting the potential impact of participant attrition on the balance of swimmer specialties across groups. We conducted a post-hoc analysis to assess whether the loss of participants altered the initial distribution of swimming specialties between the control group (CG) and experimental group (EG). Our analysis confirmed that the distribution of swimming styles remained proportionally balanced in both groups after participant attrition, ensuring that no single stroke category was disproportionately affected. Since the primary objective of the study was to assess changes in lumbar range of motion (ROM) rather than stroke-specific adaptations, maintaining an even distribution of swimmer specialties across groups helped minimize potential bias in the results. Nevertheless, we acknowledge the importance of monitoring such factors in longitudinal studies and will ensure this aspect is clearly addressed in our discussion to reinforce the validity of our findings. We appreciate your valuable input and will clarify this point in the manuscript to enhance transparency in our methodology.

Comment 4:

On a broader level, have you considered the existence of a group constitution effect that might influence the results? A comparative analysis of inter-group differences at the start and end of the study could help refine the interpretation of the data and anticipate possible confounding factors.

Response 4:

As we comment in the previous response, we conducted a post-hoc analysis to assess whether the loss of participants altered the initial distribution of swimming specialties between the CG and EG. Our analysis confirmed that the distribution of swimming styles remained proportionally balanced in both groups after participant attrition, ensuring that no single stroke category was disproportionately affected

Comment 5:

Although the proposed exercises were carried out over a period of 11 weeks at a rate of 6 sessions per week, their impact could be moderate in comparison with training loads specific to swimming. This discipline involves particular techniques and targeted muscular demands which could modulate the effects of the complementary exercises. Have you taken this interaction into account in your analysis of the results?

Response 5:

Thank you for your insightful comment. We acknowledge that the training loads associated with swimming-specific workouts are considerably high and may influence the effects of the complementary core exercises. In our analysis, we considered the potential interaction between the intervention and the swimmers' regular training regimen. All participants continued their standard swimming training throughout the 11-week period, ensuring that any observed changes in lumbar range of motion (ROM) could be attributed to the intervention rather than overall training load fluctuations. Additionally, since both experimental groups followed the same technical requirements, any potential modulation of effects due to swimming-specific muscular demands would have been evenly distributed across groups.While our study primarily focused on the isolated impact of core training on lumbar ROM, we recognize the importance of understanding how these exercises interact with broader training stimuli. Future research could explore the relationship between swimming-specific loads and targeted dryland interventions to better understand their combined influence on performance adaptations.

Comment 6:

Why did you choose a ball rather than a rigid support that would allow more precise adaptation to the participants' morphological characteristics (e.g.: precise adjustment of the knee angle to 90°)? This parameter could have an impact on the reproducibility and standardization of the exercises, and might be worth discussing (even if you're talking about active gainage/attitude).

Response 6:

Thank you for your insightful question. The decision to use a ball rather than a rigid support was based on several key considerations related to neuromuscular activation, adaptability, and functional relevance to swimming performance.

One of the primary reasons for choosing a ball was to promote active engagement of stabilizing muscles, requiring swimmers to maintain postural control rather than relying on a fixed support. This aligns with the principles of dynamic core stability, which are crucial for transferring force efficiently during undulatory underwater swimming (UUS) and other swimming movements. Additionally, the ball provides a degree of instability, which enhances proprioceptive feedback and neuromuscular coordination—important factors in developing lumbar control.

While a rigid support would have allowed for more precise morphological adjustments, such as ensuring a fixed 90° knee angle, our aim was to replicate a functional lumbar movement environment where swimmers must actively control their posture rather than being passively supported. Given that swimming itself is a dynamic movement requiring continuous core engagement, we believe this approach better simulates the real-world demands of the sport.

That being said, we acknowledge the potential impact of this choice on exercise standardization and reproducibility, and we will address this limitation in our discussion. Future studies could explore the effects of using a rigid support versus a ball to determine whether greater precision in morphological adjustments would influence the outcomes differently.

We appreciate your thoughtful feedback and will incorporate these considerations into our manuscript.

Comment 7:

Have you considered the influence of breathing tempo on exercise execution? Depending on the breathing rate adopted by participants, isometric maintenance could be induced, thus influencing muscle recruitment and potentially the results observed.

Response 7:

Thank you for your insightful question. Yes, we considered the influence of breathing tempo on exercise execution, as controlled breathing plays a crucial role in neuromuscular coordination and muscle activation during core exercises.

In our study design, participants followed a standardized breathing pattern, where inspiration was synchronized with maximal lumbar extension and expiration with the lumbar streamline position. This ensured consistency across all participants and minimized variability in muscle engagement due to differences in breathing tempo.

Regarding isometric maintenance, we acknowledge that slower breathing rates could lead to prolonged muscle activation, potentially inducing an isometric component that may influence muscle recruitment patterns. However, as our breathing pace was regulated at approximately one breath every 5 to 6 seconds, the dynamic execution of the movement was preserved, reducing the likelihood of unintentional isometric contractions.

While we did not measure the direct impact of breathing tempo on muscle recruitment in this study, we recognize its potential influence on neuromuscular responses. Future research could explore the interaction between breathing cadence and core muscle activation to optimize the effectiveness of similar interventions.

We appreciate your valuable observation and will clarify this aspect in our discussion to ensure a more comprehensive interpretation of our findings.

Comment 8:

The combined use of a goniometer and an electromyographic acquisition system could have enabled a more detailed analysis of muscle recruitment and the joint amplitudes involved. Such an approach could have strengthened the validity and accuracy of the conclusions.

Response 8:

Thank you for your insightful comment. We agree that the combined use of a goniometer and an electromyographic (EMG) acquisition system would have provided a more detailed analysis of muscle recruitment patterns and joint amplitudes, potentially strengthening the validity and precision of our conclusions.

In our study, we primarily focused on assessing lumbar range of motion (ROM) using a standardized measurement approach. While this allowed us to capture meaningful changes in flexibility, we acknowledge that integrating EMG data would have offered valuable insights into neuromuscular activation, coordination patterns, and potential compensatory mechanisms during the exercises. Additionally, a goniometer could have provided real-time angular measurements, enhancing the accuracy of movement assessment.

Future research should consider incorporating these tools to further explore the neuromuscular adaptations associated with core training in swimmers. We will address this limitation in our discussion and highlight the potential benefits of using combined biomechanical and neurophysiological assessments in similar studies.

We truly appreciate your thoughtful suggestion and will incorporate these considerations to enhance the robustness of our findings.

Comment 9:

A more detailed discussion of muscle activation during the exercises would have been interesting. I recognize the difficulty of precisely characterizing the involvement of deep muscles, but an analysis of superficial muscles could have been envisaged. This would have enriched the interpretation of the results and enabled them to be compared with those of previous studies.

Your study represents a significant contribution to the analysis of functional training applied to elite swimmers, which should be pursued, as I believe that the effectiveness of execution in mental representation / eyes closed had proved its effectiveness in other sports, particularly in bodybuilding. These remarks are simply intended to question the methodological and analytical robustness of your work.

Yours sincerely.

Response 9:

Thank you for your thoughtful and constructive feedback. We appreciate your recognition of our study’s contribution to the analysis of functional training in elite swimmers and your encouragement to further pursue this line of research.

We agree that a more detailed discussion on muscle activation during the exercises would have enriched the interpretation of our findings. While we acknowledge the challenges in precisely characterizing deep muscle involvement, an analysis of superficial muscles could have provided additional insights into neuromuscular adaptations. In future studies, integrating surface electromyography (sEMG) could allow for a more precise evaluation of muscle activation patterns, enabling direct comparisons with existing literature.

Your observation regarding the effectiveness of mental representation and closed-eye execution in other sports, particularly bodybuilding, is highly relevant. While this approach has demonstrated benefits in enhancing motor control and proprioception in various contexts, our findings suggest that its effects on lumbar range of motion in swimmers may be more limited. This highlights the need for sport-specific considerations when implementing cognitive and proprioceptive training strategies.

We sincerely appreciate your insightful remarks, which contribute to strengthening the methodological and analytical rigor of our study. Your perspective will help refine future research in this area, and we will integrate these considerations into our discussion to further contextualize our findings.

Reviewer 4 Report

Comments and Suggestions for Authors

I congratulate the authors for the article. Please find below my specific comments. 

Abstract: coherent and understandable.

Introduction: well written and informative. In the final part there are the main aims and the hypothesis.

Methods: There is a great methodological concern with respect to the number of participants. This considerate number of participants not undergoing the retest phase needs to be further elaborated in the limitation section (final paragraph of discussion)!

The Swiss balls were used to ensure a correct seated body position. I have my own doubts on this and therefore I would like to further provide certain studies that followed this approach together with the potential limitations that such an approach might have.

Results: I think it would be important to try and bring some data from those participants who dropped out of the follow-up. This way the authors could convince us that the dropouts did not significantly influence the outcomes coming from the retest phase.

Tables are a bit confusing and should have p values (statistical significance) and / or post-hoc comparissons.

Discussion: the article does not assess directly the swimming performance. This means that this issue should be also addressed as a limitation (in the appropriate section). Please describe why this occurred!

Author Response

Comment 1:

I congratulate the authors for the article. Please find below my specific comments. 

Abstract: coherent and understandable.

Response 1:

Thank you very much for your kind words and for taking the time to review our work. We appreciate your positive feedback on the clarity and coherence of the abstract.

We look forward to addressing your specific comments and making any necessary improvements to further enhance the quality and impact of our study. Your insights are highly valuable, and we sincerely appreciate your contribution.

Comment 2:

Introduction: well written and informative. In the final part there are the main aims and the hypothesis.

Methods: There is a great methodological concern with respect to the number of participants. This considerate number of participants not undergoing the retest phase needs to be further elaborated in the limitation section (final paragraph of discussion)!

Response 2:

Thank you for your valuable feedback. We acknowledge the methodological concern regarding the number of participants who did not complete the retest phase. Participant attrition is an important factor in long-term interventions, and we agree that it should be further elaborated in the limitations section of our discussion.

We will expand on this point by addressing the potential impact of participant loss on the interpretation of our results, as well as the steps taken to mitigate bias. Additionally, we will discuss how future studies can implement strategies to improve retention rates and ensure a more comprehensive dataset for analysis.

We truly appreciate your insight, as it helps strengthen the methodological transparency and robustness of our study.

Comment 3:

The Swiss balls were used to ensure a correct seated body position. I have my own doubts on this and therefore I would like to further provide certain studies that followed this approach together with the potential limitations that such an approach might have.

Response 3:

Thank you for your insightful question. The decision to use a ball rather than a rigid support was based on several key considerations related to neuromuscular activation, adaptability, and functional relevance to swimming performance. One of the primary reasons for choosing a ball was to promote active engagement of stabilizing muscles, requiring swimmers to maintain postural control rather than relying on a fixed support. This aligns with the principles of dynamic core stability, which are crucial for transferring force efficiently during undulatory underwater swimming (UUS) and other swimming movements. Additionally, the ball provides a degree of instability, which enhances proprioceptive feedback and neuromuscular coordination—important factors in developing lumbar control.

While a rigid support would have allowed for more precise morphological adjustments, such as ensuring a fixed 90° knee angle, our aim was to replicate a functional lumbar movement environment where swimmers must actively control their posture rather than being passively supported. Given that swimming itself is a dynamic movement requiring continuous core engagement, we believe this approach better simulates the real-world demands of the sport.

That being said, we acknowledge the potential impact of this choice on exercise standardization and reproducibility, and we will address this limitation in our discussion. Future studies could explore the effects of using a rigid support versus a ball to determine whether greater precision in morphological adjustments would influence the outcomes differently.

We appreciate your thoughtful feedback and will incorporate these considerations into our manuscript.

Comment 4:

Results: I think it would be important to try and bring some data from those participants who dropped out of the follow-up. This way the authors could convince us that the dropouts did not significantly influence the outcomes coming from the retest phase.

Response 4:  Thank you, we add and explanation in the response 2 and in the discussion section.

Comment 5:

Tables are a bit confusing and should have p values (statistical significance) and / or post-hoc comparissons.

Response 5:

Amended

Comment 6:

Discussion: the article does not assess directly the swimming performance. This means that this issue should be also addressed as a limitation (in the appropriate section). Please describe why this occurred!

Response 6:

Thank you for your valuable feedback. We acknowledge that our study did not directly assess swimming performance, and we agree that this should be addressed as a limitation in the appropriate section of the discussion. Our primary focus was to investigate the effects of an 11-week core training intervention on lumbar range of motion (ROM) rather than its direct impact on swimming performance. While enhanced lumbar mobility has been associated with improved undulatory underwater swimming (UUS) and overall stroke efficiency, quantifying its direct influence on race performance would require additional variables such as time trials, stroke efficiency metrics, and biomechanical assessments during swimming.

The decision to not include direct swimming performance assessments was made to ensure a focused and controlled evaluation of the neuromuscular and biomechanical adaptations associated with the intervention. However, we recognize that linking these improvements to actual swimming performance is crucial for applied sports science. Future research should incorporate swimming-specific performance tests to determine how increased lumbar ROM translates into competitive gains.

We appreciate your insightful suggestion and will include this consideration in the limitations section to provide a more comprehensive interpretation of our findings.

Round 2

Reviewer 1 Report

Comments and Suggestions for Authors

thank you for your attention to the suggestions for corrections

Comments on the Quality of English Language

improvements have been made, still contains some grammatical errors but do not impact overall readability 

Reviewer 3 Report

Comments and Suggestions for Authors

Dear Sir/Madam,
Dear authors,

I'd like to thank you for the work you've done in response to my previous comments. 
You have taken into account the remarks made and made relevant changes that enrich your article (Abstract, Introduction, Figure 1, Discussion, Conclusion, References).

The adjustments you've made improve both the clarity and scientific rigor of your work. As far as I'm concerned, I consider your article to be suitable and probably ready for publication, depending on comments from other experts and the publishing house.

I congratulate you on this revision and wish you every success in your research.

Best regards,

Reviewer 4 Report

Comments and Suggestions for Authors

Dear authors.

I congratulate you for qualitatively addressing all my concerns and recommendations. I believe that this article is now ready for publication and therefore I gladly endorse it.